Health-related quality of life and emotional and behavioral difficulties after extreme preterm birth: developmental trajectories

Vederhus Bente Johanne 1 2 3 bente.vederhus@helse-bergen.no
Eide Geir Egil 2 4
Natvig Gerd Karin 2
Markestad Trond 1 3
Graue Marit 1 5
Halvorsen Thomas 1 3
1 Department of Pediatrics, Haukeland University Hospital , Bergen , Norway
2 Department of Global Public Health and Primary Care, University of Bergen , Norway
3 Department of Clinical Science, University of Bergen , Norway
4 Centre for Clinical Research, Haukeland University Hospital , Bergen , Norway
5 Centre for Evidence Based Practice, Bergen University College , Bergen , Norway
Frydecka Dorota
Electronic publication date: 2015 Jan 20
Publication date: 2015
Volume: 3
Electronic Location ID: e738
Received 2014 Oct 17; Accepted 2015 Jan 3
Copyright: © 2015 Vederhus et al.
Copyright year: 2015
Copyright holder: Vederhus et al.
License: This is an open access article distributed under the terms of the Creative Commons Attribution License, which permits unrestricted use, distribution, reproduction and adaptation in any medium and for any purpose provided that it is properly attributed. For attribution, the original author(s), title, publication source (PeerJ) and either DOI or URL of the article must be cited.
License URL: https://creativecommons.org/licenses/by/4.0/

Keywords: Adolescent, Development, Preterm infant, Health-related quality of life, Behavior

Funding: Western Norway Regional Health Authority 390020 University of Bergen The study was supported by the Western Norway Regional Health Authority, number 390020 and the University of Bergen, Norway. The funders had no role in study design, data collection and analysis, decision to publish, or preparation of the manuscript.

==============================
Background. Knowledge of long-term health related outcomes in contemporary populations born extremely preterm (EP) is scarce. We aimed to explore developmental trajectories of health-related quality of life (HRQoL) and behavior from mid-childhood to early adulthood in extremely preterm and term-born individuals.

Methods. Subjects born at gestational age ≤28 weeks or with birth weight ≤1,000 g within a region of Norway in 1991–92 and matched term-born control subjects were assessed at 10 and 18 years. HRQoL was measured with the Child Health Questionnaire (CHQ) and behavior with the Child Behavior Checklist (CBCL), using parent assessment at both ages and self-assessment at 18 years.

Results. All eligible EP (n = 35) and control children participated at 10 years, and 31 (89%) and 29 (83%) at 18 years. At 10 years, the EP born boys were given significantly poorer scores by their parents than term-born controls on most CHQ and CBCL scales, but the differences were minor at 18 years; i.e., significant improvements had occurred in several CHQ (self-esteem, general health and parental impact-time) and CBCL (total problem, internalizing and anxious/depressed) scales. For the girls, the differences were smaller at 10 years and remained unchanged by 18 years. Emotional/behavioral difficulties at 10 years similarly predicted poorer improvement on CHQ-scales for both EP and term-born subjects at 18 years. Self-assessment of HRQoL and behavior at 18 years was similar in the EP and term-born groups on most scales.

Conclusions. HRQoL and behavior improved towards adulthood for EP born boys, while the girls remained relatively similar, and early emotional and behavioral difficulties predicted poorer development in HRQoL through adolescence. These data indicate that gender and a longitudinal perspective should be considered when addressing health and wellbeing after extremely preterm birth.

Introduction

Children born extremely preterm (EP) are at increased risk of cognitive and social shortcomings, neurosensory deficiencies, psychiatric disorders, lung problems, and cardiovascular disease (Doyle & Anderson, 2010; Hack, 2009). While early outcome studies of EP-born survivors focused mainly on those with major impairments, more attention is now being paid to consequences of less severe dysfunctions with high prevalence (Aylward, 2005) that may have unfavorable consequences for later health-related quality of life (HRQoL) and social functioning. The sparse knowledge on functional issues and HRQoL in adults born preterm is largely based on studies of subjects born in the 1970–80s, and the impression is that the majority of those without major disabilities do well and live fairly normal lives (Doyle & Anderson, 2010; Hack, 2009; Moster, Lie & Markestad, 2008; Saigal, 2013). However, neonatal intensive care went through major changes in the early 1990s, improving survival for the most immature and conceivably also the most vulnerable infants (Doyle & Anderson, 2010; Wilson-Costello et al., 2005). Thus, in Norway approximately 80% of all infants born more than 12 weeks preterm in 1999–2000 survived to discharge if resuscitated and admitted to neonatal intensive care (Markestad et al., 2005).

Emotional and behavioral problems and reduced HRQoL have been described in children born EP in the 1990s (Berbis et al., 2012; Johnson & Wolke, 2013; Stahlmann et al., 2009), but we do not know to what extent and in what ways such challenges persist to adulthood and how these issues relate to each other. Moreover, it is not known if potentially modifiable traits and characteristics may affect these outcomes, such as perceived self-efficacy. Adolescence is an important period in life, characterized by a pace in growth and development second only to that of infancy. Behavioral patterns established in this period have long-lasting effects on future health and well-being, for better or worse. In a cross-sectional study of 10 year old children born EP in 1991–92, we found that parent-reported HRQoL was reduced in boys but not in girls (Vederhus et al., 2010). We know from other studies on HRQoL and behavior of children in general that there are age-and gender-related differences which are particularly striking through puberty (Bongers et al., 2003; Michel et al., 2009). Therefore, our primary aim was to explore gender-related development of HRQoL and emotional and behavioral difficulties through puberty from 10 to 18 years of age in EP and term-born individuals, as reported by their parents. Secondly, we addressed these same issues by means of self-reports at 18 years of age, and investigated the significance of perceived self-efficacy and associations between neonatal variables and outcome at age 18.

Methods

Design and participants

A longitudinal cohort comparison design was applied. At first follow-up at 10 years of age in 2001–02, all survivors born at gestational age (GA) ≤28 weeks or with birth-weight (BW) ≤1,000 g in 1991–92 within a defined area of Western Norway Health Authority and cared for at the only neonatal intensive care unit in the region (Haukeland University Hospital) were invited (n = 35). A matched comparison group was established by inviting the temporally nearest term-born child (GA ≥ 37 weeks) of the same gender with BW between 3 and 4 kg (Norwegian 10th to 90th percentile) for each EP-born participant. If that person declined, the next was approached and so on, until a full 1:1 control group was recruited. A second follow-up was performed at 18 years of age in 2008–09. A standard clinical examination was performed by the same pediatrician (TH) at both ages and the medical history was obtained from parents or, in cases of doubt, from medical records. Perinatal characteristics of the group born EP were obtained from medical records (Table S1). Results from the first follow-up have been published (Vederhus et al., 2010).

Outcome measures—questionnaires

Participants and parents completed the questionnaires in private, either at the site of the examination or at home. Socio-demographic data were obtained from validated questionnaires used in population studies in Norway, including questions of bullying, physical activity and interaction with friends (The Nord-Trøndelag Health Study, 2012).

The Child Health Questionnaire (CHQ)

HRQoL was assessed by using the CHQ-questionnaires, which are generic instruments designed to measure functional health and well-being of children by parent or self-report (Landgraf, Abetz & Ware, 1999). The CHQ-scales address physical functioning, limitations in social functioning due to physical, emotional or behavioral constraints, general health, pain, self-esteem, mental health and behavior, impact on the parents in terms of emotional strain and time available for themselves, and significance of the child’s health on family activity and cohesion. A recall period of the preceding four weeks is used, except for family cohesion and general health. Each item is scored on a 4–6 point Likert scale, and the items in each sub-scale are summarized and transformed into a scale ranging from 0 (poor) to 100 (optimal). A validated Norwegian version was used (Selvaag et al., 2001). The parent version (CHQ-PF50) consists of 50 questions within 13 scales and was completed at 10 and 18 years. The child form (CHQ-CF87) consists of 87 questions within 12 scales and was completed by self-assessment at 18 years.

Child Behavior Checklist (CBCL) and Youth Self-Report (YSR)

Emotional and behavioral difficulties and competencies were measured using validated Norwegian versions of CBCL (parent) and YSR (self-report) (Achenbach & Rescorla, 2001; Novik, 1999). The 1991 CBCL version was completed by the parents at 10 and 18 years, while the 2001 YSR version was completed by the children at 18 years. Based on the preceding 6 months, each problem-item was scored on a scale ranging from 0 (not true) to 2 (very true or often true), making a total problem score ranging from 0–236 for the CBCL and 0–210 for the YSR. Two broadband scales were composed from five of the eight derived syndrome scales, the internalizing (withdrawn, somatic complaints, anxious/depressed) and externalizing (delinquent or rule-breaking, aggressive behavior). The competence scales measured the amount and the quality of the child’s participation in organized activity, sports and hobbies, connection with friends and family, and school performance. Higher score on each scale means more problems or better competencies.

General Self-Efficacy (GSE)

GSE denotes the perception of one’s capacity to achieve a certain goal. It reflects confidence in being able to cope across a wide range of challenging situations and seems to be important to psychosocial functioning (Schwarzer et al., 1997). A validated Norwegian 5-item version of the original 10-item GSE scale was completed at 18 years (Roysamb, Schwarzer & Jerusalem, 1999). Responses are rated from ‘not at all true’ (1) to ‘exactly true’ (4) and an overall mean value ranging from 1 to 4 is calculated.

Statistical methods

Multiple linear, binary logistic and ordinal logistic regression analyses were used to analyse continuous, binary and ordinal response variables. The mixed linear and generalized linear regression modules in SPSS were used, taking the matched structure of EP and term-born subjects (gender and age) and the repeated responses (10 and 18 years) into account. These methods allow for contribution also from pairs with declines. Separate analyses for gender were applied for the CHQ-PF50 and CBCL and as sub-analysis for the YSR. Interaction terms were used to assess differences regarding effects of time by group, and gender by group when appropriate. The regression models with self-reported outcomes were adjusted for gender, GSE, maternal education, mild mental retardation and/or ADHD. Furthermore, the regression models with parent-reported changes from 10 to 18 years in the CHQ-scales as dependent variable and the CBCL total problem score at 10 years as predictor variable were adjusted for gender, CHQ-PF50-scores and maternal education at 10 years, and mild mental retardation and/or ADHD. In the group born EP, backward stepwise linear regression analysis was applied to assess associations between outcomes at 18 years and selected neonatal variables (gender, BW < 10 percentile, intra-ventricular hemorrhage, days on mechanical ventilation, and steroid treatment for bronchopulmonary dysplasia). The parent-reported gender-related trajectories of CHQ-PF50 and CBCL were also graphically displayed.

Power analyses were originally done when the present study was launched at age 10, based on CHQ-PF50 summary scores (Vederhus et al., 2010). Applying post-hoc analyses, the study had 80% power to detect group differences of approximately 5.5 points, providing a significance level of 0.05. Findings from that study suggested that stratification on gender would provide important hypothesis-generating pilot data; hence the present design, despite the loss of statistical power. Group mean values are given with 95% confidence intervals to provide measures of uncertainty.

Ethics

The study was approved by the Regional Committee on Medical Research Ethics of the Western Norway Regional Health Authority (file no. 99.2000), and was performed in accordance with the Helsinki Declaration. Informed consent was obtained for each participating subjects and their parents.

Results

Participants

All eligible EP-born subjects (n = 35) participated at 10 years of age. On average, 1.3 term-born children were approached to recruit a full 1:1 matched control group. Clinical and socio-demographic characteristics at 10 and 18 years of age are presented in Table S1.

At 18 years, 31 (89%) of the EP and 29 (83%) of the term-born controls participated (four EP-born females and four males and two females born at term declined). Twenty-four mothers and seven fathers completed the questionnaires in the EP-born group vs. 22 mothers, five fathers and one parent couple in the control group. There were one single parent and five divorced/separated couples in both the EP and term-born group.

Development from 10 to 18 years of age as reported by parents

Results from the mixed linear regression of the CHQ-PF50-scales are presented in Table 1 and illustrated in Fig. 1. Parents of the EP-born subjects tended to give lower scores than parents of the term-born peers at both 10 and 18 years. Split by gender, the differences were smaller for girls than boys. For EP-born boys, the parents’ scores tended to improve from 10 to 18 years on all scales relative to those of the term-born controls; however, this was significant only for self-esteem, general health and parental impact-time (group by age interaction). For the EP-born girls the score trajectories did not differ from those of the controls.

Figure 1 Parent-reported changes on the Child Health Questionnaire-scales from 10 to 18 years.

Displayed for extremely preterm- and matched term-born control subjects split by gender: increasing scores imply improvements.

Table 1 Parent-reported Child Health Questionnaire at 10 and 18 years in EP- and term-born children.

Boys	Girls	
Age at follow-up:	10 years (n = 26)	18 years (n = 22)		10 years (n = 44)	18 years (n = 38a)		
CHQ-PF50 scales	Estimated mean
difference (95% CI)
EP vs. term-born	Estimated mean
difference (95% CI)
EP vs. term-born	Tests of interaction
group by age
p-valuesb	Estimated mean
difference (95% CI)
EP vs. term-born	Estimated mean
difference (95% CI)
EP vs. term-born	Tests of interaction
group by age
p-valuesb	
Physical functioning	−7.9(−15.7, −0.2)	−2.3(−10.9, 6.3)	0.33	−2.3(−6.3, 1.8)	−5.4(−9.8, −0.9)	0.31	
Role/social-
emotional/behavioral	−23.9(−42.2, −5.7)	−8.4(−28.5, 11.8)	0.25	−4.0(−11.5, 3.4)	−3.8(−11.9, 4.3)	0.96	
Role/social-physical	−10.3(−19.8, −0.7)	−4.4(−15.0, 6.2)	0.56	−0.8(−7.6, 6.1)	−1.1(−8.6, 6.4)	0.95	
Bodily pain	−8.5(−22.6, 5.7)	−6.8c(-23.1, 9.5)	0.96	−2.7(−15.2, 9.8)	−3.9c(−17.8, 9.9)	0.89	
Behavior	−24.2(−37.7, −10.6)	−7.8(−22.7, 7.2)	0.11	−6.4(−13.4, 0.6)	−4.8(−12.5, 2.9)	0.76	
Mental health	−8.8(−18.3, 0.6)	−0.5(−11.0, 9.9)	0.24	−3.9(−8.8, 1.1)	−7.0(−12.5, −1.6)	0.39	
Self-esteem	−8.3(−19.8, 3.1)	11.3(−1.6, 24.1)	0.03	−5.5(−13.6, 2.6)	−7.5(−16.4, 1.3)	0.74	
General health	−27.8(−39.2, −16.5)	−7.5(−20.2, 5.2)	0.02	−17.2(−26.3, −8.2)	−11.2(−21.2, −1.3)	0.38	
Parental impact-
emotional	−35.3(−52.9, −17.6)	−11.7(−31.3, 7.8)	0.08	−12.5(−20.3, −4.7)	−8.0(−16.6, 0.6)	0.44	
Parental impact-time	−21.4(−31.9, −10.8)	−3.8(−15.6, 8.1)	0.03	−7.1(−12.3, −1.8)	−2.1(−7.8, 3.7)	0.21	
Family activity	−23.1(−38.6, −7.5)	−2.8(−20.2, 14.6)	0.09	−6.7(−14.6, 1.2)	−5.5(−14.2, 3.2)	0.84	
Family cohesion	−2.3(−16.9, 12.3)	5.1(−11.3, 21.5)	0.50	−3.2(−13.1, 6.8)	−0.6(−11.5, 10.3)	0.73	
Notes.

EP extremely preterm

Higher scores in each of the CHQ scales reflect better outcome (range 0–100).

a One report was missing.

b From mixed linear model: interaction-term testing if development from 10 to 18 years differed in the EP vs. the term-born group.

c One missing.

Results from the mixed linear regression of the CBCL-scales are presented in Table 2 and illustrated in Fig. 2. Similarly to that of CHQ-PF50, the parents of the EP-born subjects tended to give their children poorer scores than the parents of the term-born controls on all scales at both ages, but the differences were largely greater for the boys. Relative to the controls the scores tended to improve from 10 to 18 years for the boys, although this relative improvement was significant only for the total problem, internalizing and anxious/depressed scales (group by age interaction). For the EP-born girls the score trajectories did not differ from those of the controls. Competence scores remained significantly poorer for both genders at 18 years.

Figure 2 Parent-reported changes on the Child Behavior Checklist-scales from 10 to 18 years.

Displayed for extremely preterm- and matched term-born control subjects split by gender: decreasing scores imply improvements.

Table 2 Parent-reported Child Behavior Checklist at 10 and 18 years in EP- and term-born children.

Boys	Girls	
Age at follow-up:	10 years
(n = 26)	18 years
(n = 22)		10 years
(n = 44a)	18 years
(n = 38b)		
CBCL scales	Estimated mean difference (95% CI)
EP- vs. term-born	Estimated mean difference (95% CI)
EP- vs. term-born	Tests of interaction
group by age
p-valuesc	Estimated mean difference (95% CI)
EP- vs. term-born	Estimated mean difference (95% CI)
EP- vs. term-born	Tests of interaction
group by age
p-valuesc	
Total problem	37.3 (20.7, 54.0)	9.8(−8.9, 28.5)	0.03	9.6 (2.4, 16.8)	9.8 (2.1, 17.4)	0.97	
Internalizing	10.3 (5.3, 15.3)	2.1 (−3.5, 7.7)	0.03	2.6 (−0.7, 5.8)	5.6 (2.2, 9.1)	0.20	
Externalizing	11.1(4.9, 17.3)	2.2 (−4.8, 9.2)	0.06	2.3 (−0.7, 5.2)	1.8 (−1.4, 5.0)	0.82	
Withdrawn	2.8 (1.1, 4.5)	0.3 (−1.6, 2.2)	0.06	1.3 (0.1, 2.5)	2.4 (1.2, 3.7)	0.21	
Somatic complaints	1.8 (0.4, 3.2)	0.6 (−1.0, 2.2)	0.28	0.04 (−1.3, 1.4)	0.1 (−1.4, 1.6)	0.96	
Anxious/depressed	6.2 (3.1, 9.2)	1.0 (−2.4, 4.4)	0.03	1.3 (−0.3, 3.0)	3.4 (1.7, 5.2)	0.08	
Social problems	3.9 (2.3, 5.5)	1.7 (−0.1, 3.5)	0.07	2.6 (1.6, 3.6)	1.3 (0.2, 2.4)	0.09	
Thought problems	2.2 (1.0, 3.3)	0.7 (−0.7, 2.0)	0.09	0.2 (−0.1, 0.5)	0.2 (−0.2, 0.5)	0.75	
Attention problems	6.2 (3.7, 8.7)	2.8 (−0.03, 5.6)	0.07	2.2 (0.9, 3.4)	1.2 (−0.2, 2.6)	0.30	
Delinquent behavior	2.2 (0.8, 3.7)	0.2 (−1.4, 1.8)	0.07	−0.02 (−0.8, 0.8)	0.1 (−0.7, 1.0)	0.78	
Aggressive behavior	8.8 (3.8, 13.8)	2.0 (−3.6, 7.7)	0.08	2.3 (−0.2, 4.8)	1.6 (−1.0, 4.3)	0.73	
Total competence	−3.2d (−6.1, −0.4)	−3.7e (−7.1, −0.2)	0.84	−3.8f (−5.8, −1.7)	−5.0e(−7.3, −2.8)	0.39	
Activities	−0.5g (−1.7, 0.6)	−0.8g (−2.1, 0.5)	0.78	−1.4g (−2.4, −0.3)	−1.4g (−2.5, −0.3)	0.93	
Social competence	−1.0d (−2.4, 0.5)	−1.2 (−2.8, 0.3)	0.81	−1.6g (−2.7, −0.5)	−2.0d (−3.2, −0.7)	0.66	
School competence	−2.0 (−2.9, −1.0)	−2.0h(−3.2, −0.8)	0.96	−1.0g (−1.6, −0.4)	−0.6f (−1.3, 0.0)	0.38	
Notes.

EP extremely preterm

Higher score on each of the CBCL scales reflects more problems or better competencies.

a Two reports were missing.

b One report was missing.

c Mixed linear model. Interaction-terms assessing if the differences between the EP and the matched term-born control groups were different at ten and 18 years of age.

d Two missing.

e Five missing.

f Three missing.

g One missing.

h Four missing.

Factors influencing development of HRQoL from 10 to 18 years

When adjusting for gender, ADHD/mild mental retardation, maternal education and CHQ-PF50-scores at 10 years in the mixed linear regression model, higher CBCL total problem score at 10 years predicted poorer development of CHQ-PF50 scale-scores from 10 to 18 years. This effect was statistically significant for the scales role/social-emotional/behavioral, bodily pain, mental health, self-esteem, emotional impact on the parents and family cohesion (Table 3). The effects were similar for the subjects born EP and at term. However, the effect of gender was different in the EP vs. the term-born subjects, in that EP-born boys showed significantly greater improvement in the domains mental health and self-esteem while girls showed no such improvement.

Table 3 Predictors for changes in parent reported health-related quality of life from 10 to 18 years of age for EP- and term-born children.

Response variable	Explanatory variablesa	
CHQ-PF50 scale changes
(score range 0–100)	Genderb	EP- vs. term-born
(n = 57)
b (95%CI)	CBCL total problem
score at 10 yearsc
b (95% CI)	
Physical functioning		−2.0 (−6.7, 2.7)	−0.1 (−0.2, 0.1)	
Role/social – emotional/behavioral		0.8 (−9.2, 10.9)	−0.7 (−1.0, −0.3)***	
Role/social – physical		0.9 (−6.0, 7.8)	−0.1 (−0.3, 0.1)	
Bodily paind		7.1 (−4.1, 18.3)	−0.4 (−0.7, −0.02)*	
Behavior		1.9 (−5.9, 9.6)	−0.2 (−0.5, 0.03)	
Mental health	Boys
Girls	8.5e (−1.1, 18.2)
−4.8 (−10.8, 1.2)	−0.3 (−0.5, −0.04)*	
Self-esteem	Boys
Girls	19.3f (8.9, 29.6)
−5.0 (−11.6, 1.5)	−0.3 (−0.5, −0.1)**	
General health		−2.7 (−10.0, 4.4)	−0.03(−0.2, 0.2)	
Parental impact–emotional		1.5 (−6.8, 9.8)	−0.3 (−0.6, −0.1)*	
Parental impact–time		0.9 (−2.9, 4.8)	−0.1 (−0.3, 0.01)	
Family activity		1.4 (−6.4, 9.3)	−0.2 (−0.5, 0.1)	
Family cohesion		3.5 (−6.5, 13.5)	−0.4 (−0.6, −0.1)*	
Notes.

EP = extremely preterm, CHQ = Child Health Questionnaire-Parent Form50.

a Mixed linear regression model adjusted for CHQ-scores at 10 years, gender, maternal education at 10 years and attention deficit hyperactivity disorder/mild mental retardation (n = 5). Positive b indicates improved outcome.

b Estimates are given specifically for boys and girls when there was a significant interaction group by gender.

c Child Behavior Checklist—total problem score: negative association of total problem score indicates that high level of problems at 10 years of age predicts less improvement in CHQ-scores to 18 years.

d Two missing.

e Interaction p-value = 0.017.

f Interaction p-value < 0.001.

* p ≤ 0.05.

** p ≤ 0.01.

*** p ≤ 0.001.

Self-reported CHQ and YSR at 18 years

The EP-born and term-born adolescents scored themselves similarly on CHQ-CF87, and the scores were also similar after adjusting for gender, GSE, ADHD/mild mental retardation and maternal education in regression models (Table 4). Split by gender, EP-born females had lower scores on self-esteem in the unadjusted but not in the adjusted model. For the males, there were no group differences in the unadjusted analyses, but in the adjusted model males born EP scored better than males born at term on the scales role/social emotional and self-esteem (group by gender interaction).

Table 4 Self-reports at 18 years of age in adolescents born EP and at term.

	Unadjusted results	Adjusted resultsa	
Response variable
CHQ-CF87 (range 0–100)	EP-born
(n = 31)
mean (SE)	Term-born
(n = 29)
mean (SE)	EP- vs. term-born
estimated mean differences (95% CI)	p-value	EP- vs. term-born
b (95% CI)	GSE score
b (95% CI)	Interaction term
group by gender
EP- vs. term-born
b (95% CI)	
Physical functioning	93.7 (2.9)	98.5 (0.6)	−4.4 (−9.0, 0.2)	0.06	−0.4 (−5.4, 4.7)	5, 4 (1.3, 9.6)*		
Role/social-emotional	88.9 (3.9)	92.7 (2.7)	−3.9 (−14.2, 6.3)	0.44		13.6 (6.1, 21.1)**	Boys: 17.9 (1.8, 34.0)b
Girls: −1.7 (−12.9, 9.5)	
Role/social-behavioral	95.0 (2.8)	99.2 (0.5)	−6.7 (−16.4, 2.9)	0.16	2.0 (−2.8, 6.7)	5.2 (0.6, 9.8)*		
Role/social-physical	94.6 (2.9)	100.0 (0.0)	−5.4 (−11.6, 0.8)	0.09	−0.1 (−6.4, 6.3)	8.7 (3.9, 13.6)**		
Bodily pain	74.2 (4.4)	76.6 (4.2)	−2.1 (−13.8, 9.6)	0.71	5.4 (−6.6, 17.4)	−0.4 (−10.8, 10.0)		
Behavior	76.2 (3.0)	78.2 (1.9)	−1.9 (−9.1, 5.3)	0.59	4.6 (−2.6, 11.8)	9.9 (3.9, 15.8)**		
Mental health	72.9 (3.2)	75.1 (2.1)	−2.1 (−10.1, 5.8)	0.59	2.5 (−5.9, 10.9)	10.6 (4.1, 17.1)**		
Self-esteem	69.7 (2.9)d	75.4 (2.6)	−6.1 (−14.4, 2.3)	0.15		15.1 (9.4, 20.9)***	Boys: 14.2 (3.4, 25)c
Girls: −5.2 (−12.7, 2.2)	
General health	70.4 (3.3)	73.7 (3.1)	−3.0 (−11.6, 5.6)	0.49	3.7 (−5.9, 13.3)	11.6 (3.8, 19.5)**		
Family activity	88.0 (3.3)	91.4 (2.5)	−3.8 (−13.1, 5.5)	0.40	1.0 (−8.6, 10.6)	3.4 (−3.8, 10.7)		
Family cohesion	74.5 (4.9)	74.5 (4.7)	0.1 (−13.9, 14.0)	0.99	7.6 (−7.5, 22.7)	15.1 (2.7, 27.5)*		
YSR:								
Total problems
(range 0–210)	33.0 (5.2)	29.1 (3.1)	3.7 (−8.7, 16.1)	0.54	−9.9 (−21.6, 1.9)	−20.7 (−30.3, −11.1)***		
Internalizing
(range 0–62)	10.2 (2.0)	8.2 (1.2)	1.9 (−2.8, 6.6)	0.41	−2.9 (−7.4, 1.5)	−9.4 (−12.8, −6.0)***		
Externalizing
(range 0–64)	8.4 (1.3)	9.8 (1.0)	−1.4 (−4.7, 1.9)	0.39	−3.9 (−7.4, −0.5)*	−3.3 (−6.1, −0.4)*		
Notes.

EP extremely preterm

CHQ Child Health Questionnaire-Child Form87

YSR Youth Self-Report

GSE General Self-Efficacy (score range 1–4)

Higher scores reflect better outcome on the CHQ, but more problems on the YSR.

a Mixed linear model adjusted for gender, maternal education at 10 years of age, GSE and attention deficit hyperactivity disorder/mild mental retardation. If interaction term group by gender was not significant (p > 0.05) model without is reported.

b Test of interaction assessing if differences between boys and girls were different in the EP and the term-born group, p = 0.040.

c Test of interaction assessing if differences between boys and girls were different in the EP and the term-born group, p = 0.003.

d Split by gender, EP female 66.6 vs. term-born female 78.8, p-value = 0.009.

* p < 0.05.

** p < 0.01.

*** p < 0.001.

On the YSR, the EP and term-born males scored themselves similarly while the EP-born females gave poorer scores than term-born females on the withdrawn and anxious/depressed scales (1.5; 95% CI [−0.0–3.1]; p = 0.05 and 2.7; 95% CI [0.2–5.1]; p = 0.03, respectively) and the social competence scale (−1.6; 95% CI [−2.7–−0.5]; p = 0.008). In the adjusted model, EP-born subjects reported significantly fewer externalizing problems than their term-born peers (Table 4).

GSE score was positively associated with most of the CHQ-CF87-scales and negatively associated with the internalizing, externalizing and total problem scale on the YSR in both EP and term-born adolescents (Table 4).

Influence of neonatal variables in EP-born subjects at 18 years

In the backward stepwise linear regression analysis, only the number of days of mechanical ventilation remained a significant predictor of the three CBCL scales social, thought and attention; b = 0.09; 95% CI [0.03–0.15] (adjusted R2 = 0.22), b = 0.06; 95% CI [0.03–0.09] (adjusted R2 = 0.29), and b = 0.16; 95% CI [0.07–0.25] (adjusted R2 = 0.28), respectively. No significant associations were found with any of the CHQ-scales.

Discussion

To our knowledge, this is the first controlled study reporting longitudinal parent assessments of HRQoL and emotional and behavioral difficulties from mid-childhood to early adulthood in subjects born EP in the 1990s. At 10 years of age, boys born EP were assessed as having substantially more problems with emotion/behavior, competence and HRQoL than peers born at term, while at 18 these differences were considerably smaller. For the girls, differences were generally smaller and did not change much during puberty. Emotional and behavioral difficulties at 10 years of age were associated with less favorable development of HRQoL through puberty in both the EP and the term-born group. However, judged by self-reports at 18 years, HRQoL and emotional/behavioral issues were scored relatively similar in the EP and term-born groups, except that EP-born girls reported more withdrawal, anxiety and depression and lower social competence. GSE was associated with better HRQoL and less emotional and behavioral difficulties irrespective of group.

Developmental trajectories

The longitudinal data obtained from the parents suggest that negative consequences of preterm birth on HRQoL and behavior tend to diminish over time for boys. A positive development of HRQoL corresponds with data presented in a review article, but in that paper there was discussion on whether improvements reflected a shift from parent to self-report, which is known to provide different assessments (Zwicker & Harris, 2008). Contrary to our findings, a previous study of EP-born and term-born subjects born in 1977–82 reported that all participants perceived a small decrease in HRQoL from adolescence to young adulthood, but that was by self-reports (Saigal et al., 2006). A decline in general health was also reported by children born with extremely low birth-weight in 1992–95 and their term-born controls from 8 to 14 years of age (Hack et al., 2012). The difference in HRQoL and behavior trajectories between EP-born boys and girls observed in our study may be due to the fact that girls were assessed as having fewer difficulties at 10 years and therefore had less room for improvement. Also to be considered is that girls, in general, tend to have a less favorable development of HRQoL and mental health through puberty than boys (Bongers et al., 2003; Michel et al., 2009). There were, however, still differences in scores between EP and term-born subjects of both genders at 18 years. It is therefore likely that some differences in HRQoL and behavior will remain towards young adulthood. Regarding the significance of gender, studies of EP-born subjects have been inconsistent in that more emotional/behavioral difficulties have been described for males at 17 years (Grunau, Whitfield & Fay, 2004) as well as for females at 20 years (Hack et al., 2004). Internalizing symptoms in eight year old EP-born girls predicted persistent internalizing symptoms at 20 years in a study by Hack et al. (2005), which is in accord with our observation.

Normal behavior and HRQoL are important premises for social functioning, and the improvements towards that of peers born at term at young adulthood are reassuring in terms of the likelihood of living normal self-supportive lives. This notion is supported by a register-based study of adults in Norway which, after excluding people on disability pensions, demonstrated that level of education and chance of having a paid job only marginally decreased with degree of prematurity (Moster, Lie & Markestad, 2008). However, these adults were born during 1967–83 when few EP-born infants survived.

Inferior HRQoL has been reported in children and adolescents with behavioral problems (Klassen, Miller & Fine, 2004) as well as in children born EP (Saigal et al., 2006; Verrips et al., 2012). In one study, internalizing behavior at eight years predicted poorer HRQoL at 13 years in children born with very low birth-weight or very preterm, but not in the term-born control children (Wolke et al., 2013). In contrast, we found that emotional/behavioral difficulties at 10 years predicted less improvement in HRQoL at young adulthood both for EP-born and term-born subjects, suggesting that the effect of behavior is a general phenomenon and not related to prematurity per se.

Self-assessment at 18 years

The finding that EP and term-born individuals scored their HRQoL similarly is in agreement with most reports on survivors of preterm birth (Hack et al., 2007; Indredavik et al., 2005; Roberts et al., 2013; Saigal et al., 2006), but in contrast to some who report inferior mental health and well-being (Lund et al., 2012; Methusalemsdottir et al., 2013; Saigal et al., 2007). Earlier reports on behavior are more inconsistent (Boyle et al., 2011; Dahl et al., 2006; Grunau, Whitfield & Fay, 2004; Hack et al., 2004; Lund et al., 2012; Saigal et al., 2003). Interestingly, the EP-born males reported their self-esteem to be better than the term-born males, whereas the scores were opposite for the females. This gender difference was in agreement with their parents’ assessments. Others have also reported that self-esteem of EP-born adolescents was comparable to that of controls, but not the gender difference observed in the present study (Roberts et al., 2013; Saigal et al., 2002).

Contrary to a previous study (Verrips et al., 2012), we found that confidence in the ability to achieve goals (GSE score) was positively associated with self-reported HRQoL and emotional health irrespective of group. This finding may be important because GSE is a trait that can be modified and strengthened by interventions (Axelsson et al., 2013). Moreover, it was encouraging that EP-born subjects reported fairly similar levels of leisure time activities and social interaction as term-born peers, which is contrary to previous reports of disadvantages in sports and friendship (Dahan-Oliel, Mazer & Majnemer, 2012; Hallin & Stjernqvist, 2010). Bullying in preterm born adolescents have been reported (Grindvik et al., 2009), but was not found in the present study.

Little is known concerning the significance of neonatal exposures on later HRQoL and emotional issues. Neonatal morphine exposure and neonatal procedural pain predicted internalizing behavior in 7-year old very preterm born children in one study (Ranger et al., 2013). We found an association between duration of mechanical ventilation and emotional problems at 18 years. We have previously reported on a negative association between neonatal corticosteroid treatment and social functioning at ten years (Vederhus et al., 2010), but no such association was found at 18 years.

Strengths and weaknesses

The major strengths of this study were the population-based and controlled design and the high follow-up rate, and that the assessments were made by the same research team on both occasions. No subjects in this birth-cohort had major impairments, and there were no exclusions for medical reasons, thus the results may not be generalized to those with severe neurodevelopmental impairments. The use of both parent- and self-reports facilitated better understanding of developmental patterns.

The limitation of this study was primarily a relatively low number of participants, and positive as well as negative findings must therefore be cautiously interpreted. HRQoL research is characterized by a high number of endpoints, challenging a priori power calculations, which was further complicated in this study by partly unknown distributions in preterm born populations, and by the longitudinal perspective. Although debatable (Fayers & Machin, 2007), a post hoc power analysis was reported at age 10 (Vederhus et al., 2010), indicating a between-group detection limit of 5.5 for CHQ-PF50 summary scores, a figure considered adequate by the creator (Landgraf, Abetz & Ware, 1999). Stratifying by gender reduced the size of the groups, and therefore also statistical power. Nevertheless, this was considered important and necessary, given the findings at age 10 and the known influence from gender during puberty on the issues under study. Given the relatively small sample size, the present findings should primarily be considered hypothesis-generating pilot data and an incitement for future research, preferably in larger population samples. To a large extent we have given 95% confidence intervals for calculated group mean values in order to indicate the reliability of the estimates.

Conclusion

For EP-born boys, the parents’ assessments of HRQoL, emotion and behavior improved towards that of term-born controls from 10 to 18 years of age, and for both genders and for children born EP and at term emotional/behavioral difficulties at 10 predicted less improvement in HRQoL at age 18. As there were still some important group differences at age 18, we are concerned that our findings may herald difficulties in a complex and demanding adult life. The present pilot data imply that gender and a longitudinal perspective are important aspects when addressing HRQoL and behavioral issues after preterm birth, and that these issues should be considered in clinical work as well as in future and preferably larger studies.

Supplemental Information

Table S1 Clinical, socio-demographic and neonatal characteristics of the extremely preterm(EP) and term-born children

Click here for additional data file.

We sincerely thank the children and their parents who gave of their time to take part in this follow-up study.

Additional Information and Declarations

Competing Interests

Author Contributions

Human Ethics

The authors declare there are no competing interests.

Bente Johanne Vederhus conceived and designed the experiments, performed the experiments, analyzed the data, wrote the paper, prepared figures and/or tables, reviewed drafts of the paper.

Geir Egil Eide analyzed the data, prepared figures and/or tables, reviewed drafts of the paper.

Gerd Karin Natvig analyzed the data, reviewed drafts of the paper.

Trond Markestad conceived and designed the experiments, analyzed the data, wrote the paper, reviewed drafts of the paper.

Marit Graue conceived and designed the experiments, analyzed the data, reviewed drafts of the paper.

Thomas Halvorsen conceived and designed the experiments, performed the experiments, analyzed the data, wrote the paper, reviewed drafts of the paper.

The following information was supplied relating to ethical approvals (i.e., approving body and any reference numbers):

The Regional Ethics Committee, University of Bergen approved the study (file no. 99.2000), that was performed in accordance with the Helsinki Declaration of 1975, as revised in 1983.

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
