# Peer review of "Health-related quality of life and emotional and behavioral difficulties after extreme preterm birth: developmental trajectories"

_PeerJ, doi:10.7717/peerj.738_

## Round 0.1 · original submission · Minor Revisions

In summary, this is a well done study that unfortunately suffers from small sample size and therefore should be considered pilot data only. This should be clearly stated in any publication.

Reviewer 1 ·

Basic reporting

For very preterm infants, quality of life across childhood and into early adulthood has received very little attention. This longitudinal study investigates HRQOL in a cohort of infants born at < 1000 grams or < 28 weeks gestational age at 10 and 18 years of age. Specific attention is paid to emotional and behavioral problems.

The paper is very well written,

Experimental design

Valid measurement tools were used, and the statistical analysis is appropriate, however the sample size is too small from which to draw conclusions with any confidence.

Although the study was performed with a population- based sample some discussion in the methods of the power of the study to detect real differences in outcomes is needed.

he control group is small given the 1:1 matching design so a comment on whether the control group is representative of Norwegian children is the region would strengthen the comparison. A control group of 2 or 3:1 would have been preferable, although more costly. Subgroup analysis by gender compounds the problem of small sample size but was the primary aim of the study and yields interesting pilot data for a larger study.

The results are communicated with many tables and graphs, but I found they conveyed important information that I wanted to know.

Validity of the findings

The weaknesses of the study are appropriately acknowledged and discussed in the Discussion section of the paper.

In summary, this is a well done study that unfortunately suffers from small sample size and therefore should be considered pilot data only. This should be clearly stated in any publication.

Reviewer 2 ·

Basic reporting

The authors report on a longitudinal study of 18 year olds born very preterm (<29 weeks GA) or very low birth weight (<1000g) along with who were also examined at age 10 years born in their NICU over 1990/1. Their primary aim was to examine HRQoL and emotional and behavioural problems over these years in a contemporary cohort and investigate the significance of self efficacy and neonatal variables in these outcomes. A control group was recruited at age 10 years and also followed up at age 18 years.
Their study of a contemporary cohort of young people born in the post surfactant era where there is marked improvement in NICU outcomes is important as there are few studies to date looking at emotional and behavioural outcomes and none looking at self reported quality of life as teens.
They show clearly the different trajectories for boys compared with girls and how these change over puberty. Their line graphs are a particularly powerful way to represent these changes and should be more prominently displayed, perhaps with some better graphics ? using colour to display the changes over time more succinctly than the whole page of mini graphs.

Experimental design

No comments

Validity of the findings

The small number of participants (N=35) limits the fine grain analysis of their large data set and I believe that they should concentrate on the main findings and not obscure these by over reporting of other less secure data due to power issues.

Additional comments

Overall this is a well designed and well written paper with appropriate references to the extant literature.

---

## Round 0.2 · accepted · Accept

Your manuscript is an important contribution to the field.